# Prevalence and associated factors of suicidal behavior among adolescents in Bahir Dar City, Northwest Ethiopia

Belete Birhan[1], Michael Beka[2], Habte Belete[2], Abdulbasit Sherfa[3]*, Gidey Rtbey[4]

1 Department of Psychiatry, College of Medicine and Health Sciences, Wolayita Sodo University, Wolayita Sodo, Ethiopia, 2 Departments of Psychiatry, College of Medicine and Health Sciences, Bahir Dar University, Bahir Dar, Ethiopia, 3 Department of Public Health, College of Medicine and Health Sciences, Werabe University, Werabe, Ethiopia, 4 Department of Psychiatry, College of Medicine and Health Sciences, University of Gondar, Gondar, Ethiopia

* abdusherfa3@gmail.com

## Abstract

### Background

Suicide is a significant public health issue worldwide and the third leading cause of death among 15–19-year-olds. In Ethiopia, while suicidal behaviors among adolescents have a considerable impact, there is limited information on these behaviors, especially in community settings. The study aimed to assess the prevalence and associated factors of suicidal behavior among adolescents in Bahir Dar City.

### Methods

A community-based cross-sectional study involving 692 adolescents was conducted in Bahir Dar City, Northwest Ethiopia, in May 2021. Participants were selected through a multistage sampling technique. Semi-structured and standardized questionnaires were employed to gather data via face-to-face interviews. Suicidal behavior was evaluated using the six questions of the Mini International Neuropsychiatric Interview. Epi-Data version 3.1 was utilized for data entry, whereas Statistical Package for Social Science version 24 was employed for analysis. Logistic regression with an adjusted odds ratio determined the relationship between outcome and factor variables. A p-value of < 0.05 was considered statistically significant.

### Results

The prevalence of suicidal behavior was found to be 19.8%, with a 95% CI (16.6, 22.8). Regarding predictive variables, low social support (AOR 12.42, 95% CI 5.95 to 25.95), expressed more anger (AOR 7.30, 95% CI 2.51 to 21.22), negative childhood experiences (AOR 7.11, 95% CI 3.56 to 14.20), family history of suicidal attempts (AOR 5.66, 95% CI 2.44 to 13.10), having no good relationship with parents (AOR

**Data availability statement:** All relevant data are within the paper and its Supporting Information files.

**Funding:** The author(s) received no specific funding for this work.

**Competing interests:** The authors have declared that no competing interests exist.

**Abbreviations:** ACE, Adverse Childhood Experience; AORs, Adjusted Odds Ratios; BSRS, Bergen Social Relationships Scale; LMICs, Lower and Middle Income Countries; MINI, Mini-International Neuropsychiatric Interview; PHQ-9, Patient Health Questionnaire-9; PI, Principal Investigator; RSES, Rosenberg Self-Esteem Scale; SLESQ, Stressful Life Events Screening Questionnaire; SPSS, Statistical Package for Social Science; WHO, World Health Organization.

5.59, 95% CI 2.59 to 12.08), depression (AOR 5.32, 95% CI 2.64 to 10.70), stressful life events (AOR 2.91, 95% CI 1.45 to 4.91), and interpersonal stress (AOR 2.17, 95% CI 1.07 to 4.40) were significantly increased odds of suicidal behavior among respondents.

## Conclusion

Suicidal behavior was prevalent in the study area, and factors such as depression, stressful life events, poor parental relationships, negative childhood experiences, high anger expression, low social support, and a family history of suicidal attempts were the significant contributors. Mental health interventions have to emphasize psychosocial support for adolescents, improve the parent-adolescent relationship by increasing parental knowledge about adolescents' mental health, and integrate youth-friendly health services that foster mental well-being. Policymakers can also propose mandating mental health guidance in schools.

## Introduction

Suicide is death caused by an act of self-harm that is intended to be lethal. Suicidal behavior includes completed suicide, attempted suicide, and suicidal ideation, which provides for suicide-related thoughts, plans, and acts [1]. Suicidal behaviors are a serious global public health issue, accounting for more than 2.4% of the global disease burden by 2020 [2]. An estimated 703,000 people died by suicide worldwide in 2019, with low- and middle-income countries (LMICs) accounting for roughly three-fourths of all deaths. Suicide was the 4th and the 3rd leading cause of death for those aged 15–29 and 15–19, respectively, and nearly 9 out of 10 were from LMICs, which are home to nearly 90% of all adolescents worldwide [3–6]. Africa has six of the ten highest suicide rates in the world, with 11 suicide fatalities per 100,000 people annually [7]. According to the World Health Organization (WHO, 2021), about 34,000 suicides are reported in Africa each year, accounting for approximately 3.2 per 100,000 people [8,9]. According to pooled data, the prevalence of suicidal ideation and attempted suicide in Ethiopia ranged from 1% to 55% and 0.6% to 14% [10].

Adolescence is a critical period marked by physical, emotional, and societal changes, in which people might engage in a variety of risky behaviors, including suicidal ideation, suicide attempts, and suicide [11–13]. Suicide is a principal cause of death among adolescents. Self-harm is the most significant risk factor for suicide, yet the majority of self-harm does not come to the attention of health services [14]. Suicidal behavior, deliberate self-harm, and non-suicidal self-injury are significant precursors of suicide in children and adolescents [15]. Adolescent psychiatric emergencies are a leading cause, with suicidal attempts and deaths being the strongest predictors [16]. Suicidal thoughts and suicidal behaviors develop during adolescence and peak late in adolescence and early adulthood [17].

Worldwide, the possible risk factors of suicide include victimization (bullying, sexual harassment), poor psychological state related to depression, phobic disorders,

anxiety, alcohol use disorder, child abuse, impulsivity, and lack of parental understanding [12,18,19]. Studies show that mental state issues, substance use, genetic and biological factors, poor physical health, physical disability, and family-environmental factors increase the risk of suicidal behaviors in adolescents [16,20,21]. Suicide is a prevalent issue among adolescents, with those who have experienced suicidal ideation and attempts being significantly more likely to commit suicide [22].

Suicide costs billions of dollars in lost productivity and healthcare expenditures, requiring comprehensive prevention initiatives, enhanced mental health services, public awareness campaigns, and community engagement [23]. Suicide prevention requires robust healthcare systems, universal health coverage, and community engagement. WHO's toolkit aims to address stigma, improve knowledge, and provide social support [24]. In 2013, the 66th World Health Assembly adopted the WHO's Mental Health Action Plan, focusing on suicide prevention to reduce suicide rates by 10% by 2020 [25].

Adolescent suicidal behavior is a neglected public health issue, especially in LMICs, including Ethiopia [26]. Underinvestment in mental health and lack of national health insurance coverage in LMICs make it more difficult to get resources and help, particularly for young people between the ages of 15 and 29, who are faced with trauma, social stigma, and financial difficulties [27]. It is a multipart and multidimensional phenomenon stemming from the interaction of several factors [28].

Suicide rates are increasing rapidly among young individuals, particularly males aged 15–24, and the rate is still growing [10]. Even though LMICs account for the bulk of suicides, high-income Western nations are the source of most of the knowledge about suicidal behaviors. Because there aren't many studies on teenagers in Ethiopia, little is known regarding the prevalence and determinants of suicidal behavior (suicidal thoughts, suicidal plans, and suicidal attempts). Therefore, this study aimed to evaluate the prevalence of suicidal behavior and its contributing factors among adolescents.

## Methods and material

### Study setting

The study was conducted in Bahir Dar City, Northwest Ethiopia, which is 565 kilometers from Addis Ababa. According to the Bahir Dar City mayor's office report in 2019/2020, the estimated population in the city was 373,073. From the total population, the number of adolescents (between 10–19 years) was 84,382. There are three public hospitals in the city: one primary, one specialized, and one comprehensive. While specialized hospitals concentrate on a single area of healthcare and treat cases within their field, comprehensive specialized hospitals in Ethiopia offer a wider variety of advanced services across many specializations and handle more difficult and diversified cases. In addition, the city has four private hospitals. Inpatient psychiatric care is only offered by comprehensive and specialized public institutions [29,30].

### Study design and period

In May 2021, a community-based cross-sectional survey was carried out in Bahir Dar City, Northwest Ethiopia. The method was used to assess the prevalence of suicidal behaviors and associated factors in the population at a specific point in time. Cross-sectional studies are frequently employed in public health research to assess the prevalence of health problems and their factors [31].

### Study participants

All adolescents (aged 10–19 years old) residing in Bahir Dar City were the source population, whereas adolescents who had been living for the last six months within the selected sub-cities of the town were the study population. Moreover, households were taken as the sampling unit, and the selected eligible adolescent was considered the study unit. Adolescents who had been residing for the last six months in Bahir Dar City were included, and adolescents from selected households who were severely ill and unable to respond to the interviewer properly were excluded.

**Sample size and sampling methods.** The sample size was determined with a single population proportion formula based on a previously reported suicide behavior prevalence of 22.5% from a study done in Ethiopia [32]. A 95% confidence interval, a 4% margin of error, and a design effect of 1.5 were employed.

The sample size was calculated as: $N = Z\alpha/2 \times P(1-P)/W2$.

Where $Z\alpha/2 = confidence\ level\ (1.96)\ at\ CI\ of\ 95$

N = Sample size

P = 22.5% (proportion of adolescents who have suicidal ideation)

W = margin of sampling error.

1- P = sample error proportion of adolescents who have no suicidal ideation

N = $(1.96)^2$ (0.225) (1–0.225)/ $(0.04)^2$ = 419, then multiply by 1.5 for design effect, which gives 629. By assuming a 10% non-response rate, the final sample size was 692.

The multistage sampling method was employed to choose three sub-cities from a total of six and to choose six corresponding administrative kebeles (the smallest administrative entity) from a total of seventeen. The households in the administrative kebeles were selected by systematic random sampling technique after identifying an initial starting household by use of a random number. Eligible adolescents in the selected households were further selected and interviewed. Only one adolescent member of the household was selected by the lottery method for the interview on suicide behavior.

**Data collection methods and tools.** The questionnaire included items to assess socio-demographic information, suicidal behavior, self-esteem, social support, adverse childhood experience, anger expression, stressful life events, clinical factors, behavioral factors, and interpersonal stress.

To ensure the data quality, the questionnaire was first developed in English, translated into Amharic, and translated back into English by different experts to check its consistency. The data was collected using an interviewer-administered questionnaire from the selected household by three health extension workers, one BSc psychiatry nurse supervisor, and the principal investigator, who was involved in the supervision.

The principal investigator trained the data collectors and supervisors for one day. A pre-test was carried out on 5% of the respondents (35 adolescents) in Woreta town, and according to the pre-test, the questionnaire did not need to be modified. The supervisor and principal investigator closely followed the daily data collection process for completeness, clarity, and consistency.

## Data collection tools

**Suicidal behaviors.** Participants who answered "yes" to at least one of the six questions in the Mini-International Neuropsychiatric Interview (MINI) were categorized as displaying suicidal behavior. The MINI shows high internal consistency (Cronbach's alpha = 0.84) [33].

**Social support.** The participants' social support was assessed using the Oslo-3 social support scale [25]. Based on their scores, participants were classified as follows: those with scores between "3–8" were considered to have poor social support, scores of "9–11" indicated moderate support, and scores ranging from "12-14" represented strong social support.

**Depression.** Depression was assessed by a nine-item Patient Health Questionnaire-9 (PHQ-9), which has four response categories referring to the amount of time that the symptom was present (not at all (0), several days (1), more than half of the days (2), nearly every day (3)), with a total score ranging from 0 to 27 [34]. Adolescents who scored ≥5 in PHQ-9 were considered as having depression [35].

**Adverse child hood experience.** The ACE Questionnaire (Adverse Childhood Experiences Questionnaire) was used to assess the impact of traumatic childhood experiences, including abuse, neglect, and household dysfunction. It consists of questions about various adverse experiences [36].

**Self-esteem.** The participant's self-esteem was assessed using the Rosenberg Self-Esteem Scale (RSES). People scoring between 15 and 25 are average. A score of less than 15 suggests low self-esteem, and a score greater than 25 suggests high self-esteem [37].

**Anger expression.** The participants' expression of anger was evaluated using the Spielberger Anger-Out Expression Scale (SAOES), which includes eight items designed to assess the coping strategies individuals employ, especially in relation to outward expressions of anger. This scale categorizes anger expression into three levels: low anger expression (scores below 10), moderate anger expression, and high anger expression (scores of 15 or above) [38].

**Stress full life events.** According to the SLESQ (Stressful Life Events Screening Questionnaire) definition, individuals who reported a positive response to at least one of the eleven specific categories or the two general categories of events—such as enduring a life-threatening accident, experiencing physical or sexual abuse, or witnessing a murder or assault—were classified as having undergone stressful life events [39].

**Interpersonal stress.** Interpersonal stress was assessed by using the Bergen Social Relationships Scale (BSRS). The scale consists of six items. Individuals with high scores (above the mean) on BSRS have high interpersonal stress [40].

**Fear of corona virus.** The assessment of fear associated with the coronavirus was carried out using a specially designed seven-item scale. Participants indicated their level of agreement with a range of statements using a five-point Likert scale, which included the options "strongly disagree," "disagree," "neutral," "agree," and "strongly agree." Individuals with high scores (above the mean) have a fear of the coronavirus [41]. Clinical variables like family history of suicidal attempts and other factors like growing up with their parent were assessed by asking the participants.

## Data analysis

The data was entered into Epi-Data version 3.1 and exported to SPSS version 24 for analysis. Necessary data processing like recoding, categorizing, merging, computing, and counting was done before the actual data analysis. After data processing, descriptive statistics, like measures of central tendency (mean, median, and mode) and measures of dispersion, were used for continuous variables, and frequency count and proportion were used to summarize categorical variables. All variables with a p-value of less than 0.2 in the bivariate logistic regression analysis were entered into the multivariable logistic regression model to identify factors associated with suicidal behavior. The adjusted odds ratios (AORs) with 95% confidence intervals were used to assess the strength of associations between the outcome and predictor variables. The p-value of $< 0.05$ was considered significant.

## Ethical considerations

Ethical clearance was obtained from the Institutional Review Board (IRB) of Bahir Dar University, College of Medicine and Health Science [REF.178/2021]. Then a letter of permission was obtained from the Bahir Dar city administration health office, and permission was granted from the city administrator. The aims of the study were explained to the participants, and data were collected after obtaining assent and written consent. Adolescents aged 18 years and above provided informed written consent to participate. Participants under the age of 18 gave verbal assent to the study, and written consent was then obtained from their parents or guardians on their behalf. Both adolescents and parents were informed that they had the right to refuse to answer any question at any time. Referral was given for participants who had suicidal behavior and depressive symptoms.

## Results

### Socio demographic characteristics of the respondents

A total of 631 adolescents participated in this study, yielding a response rate of 91.2%. The average age of the respondents was 16.1 years (SD = 2.0), with 99.4% being unmarried. Most respondents (83.7%) resided with their parents, while 6.5% lived with others, and 9.8% had independent living situations. Over half of the respondents (55.2%) identified as female (Table 1).

## Psychosocial and substance related factors

The study found that 13.5% of respondents had a family history of suicide attempts, while only 0.3% indicated a family history of completed suicide. Almost 80% of the respondents were raised by their biological parents. Additionally, 14.1% (89 individuals) characterized their family relationships as poor. In contrast, most respondents (99.7%) reported having positive relationships with their peers. Many respondents indicated experiencing moderate levels of social support. The rates of tobacco, alcohol, and chat usage among respondents were recorded at 4.4%, 9.2%, and 5.1%, respectively (Table 2).

## Clinical and other related factors

Out of the total respondents, 121 individuals, or 19.2%, were found to be experiencing depression. Regarding stressful life events, 40.9% of respondents reported having encountered such situations. Additionally, more than one-third (36.3%) indicated that they had faced adverse childhood experiences. Concerning self-esteem, 24.6% of respondents reported low self-esteem, while 60.4% reported moderate self-esteem, and 6% reported high self-esteem. Moreover, 62.4% of respondents displayed moderate anger expression behaviors, with 13.8% showing higher levels of anger expression. Approximately 60% of respondents expressed a fear of COVID-19, and 51.5% reported experiencing high levels of interpersonal stress.

## The magnitude of suicidal behavior

The study revealed a prevalence of suicidal behavior at 19.8% (125 individuals), with a 95% confidence interval ranging from 16.6 to 22.8. Almost half (48%) of the respondents were late adolescents (ages 18–19), and a significant majority (71.2%) were female. Among female respondents, 25.6% reported experiencing suicidal behavior, while 74.4% indicated they had not. Furthermore, 34% reported a family history of suicide attempts, 85% had experienced adverse childhood events, and 22.4% had a history of alcohol use. Almost half (48%) indicated poor relationships with their parents, while 76% reported experiencing stressful life events. Regarding anger expression, 53.6% of participants exhibited moderate levels, while 33.6% displayed high levels. Among those with low anger expression, 89.3% indicated they had not engaged

Table 1. Socio-demographic characteristics of participants.

| Variable | Category | Frequency | Percentage |
|---|---|---|---|
| Sex | Female | 348 | 55.2 |
| | Male | 283 | 44.8 |
| Age | Early adolescent(10–13) | 75 | 11.9 |
| | Middle adolescent(14–17) | 373 | 59.1 |
| | Late adolescent(18–19) | 183 | 29 |
| Marital status | Single | 627 | 99.4 |
| | Married | 2 | 0.3 |
| | Divorced | 2 | 0.3 |
| Religion | Orthodox | 509 | 80.67 |
| | Muslim | 102 | 16.16 |
| | Protestant | 20 | 3.17 |
| Living status | With family | 528 | 83.7 |
| | Alone | 62 | 9.8 |
| | Other | 41 | 6.5 |
| Educational level | No formal education | 2 | 0.3 |
| | Primary school | 299 | 47.4 |
| | High school and above | 330 | 52.3 |

*other (participants live with grandmother, grandfather, aunt, uncle).

**Table 2. The frequency distribution of participants' psychosocial and substance-related factors.**

| Variable | Category | Frequency | Percentage |
|---|---|---|---|
| Family history of suicide attempt | Yes<br>No | 85<br>546 | 13.5<br>86.5 |
| Family history of suicidal committed | Yes<br>No | 2<br>629 | 0.3<br>99.7 |
| Grow up with your biological family | Yes<br>No | 503<br>128 | 79.7<br>20.3 |
| Relationship with parent | Good to very good<br>Not good/disturbed or very disturbed | 542<br>89 | 85.9<br>14.1 |
| Relationship with peers | Good<br>Poor | 629<br>2 | 99.7<br>0.3 |
| Social support | Poor<br>Moderate<br>Strong | 251<br>300<br>80 | 39.8<br>47.5<br>12.7 |
| Ever used tobacco/ cigarettes | Yes<br>No | 28<br>603 | 4.4<br>95.6 |
| In the past three month have you used tobacco/cigarettes | Yes<br>No | 23<br>608 | 3.6<br>96.4 |
| In your life, have you Ever used alcohol | Yes<br>No | 58<br>573 | 9.2<br>90.8 |
| in the past three month have you used alcohol | Yes<br>No | 50<br>581 | 7.9<br>92.1 |
| Ever use of khat | Yes<br>No | 32<br>599 | 5.1<br>94.9 |
| Khat use in the past three month | Yes<br>No | 26<br>605 | 4.1<br>95.9 |

in suicidal behavior, whereas 10.7% did report such behavior. In comparison, only 51.7% of individuals with high anger expression reported no suicidal behavior, with 48.3% admitting to it (Fig 1).

## Factors associated with suicidal behavior among adolescents

In the multivariate analyses, family history of suicidal attempts, facing stressful life experiences, having adverse childhood experiences, having poor social support, not having a good relationship with their parents, depression, interpersonal stress, and having high anger expression behaviors were found to be significantly associated with suicidal behavior at $p < 0.05$.

Adolescents with poor social support were approximately 12.42 times more likely (AOR = 12.42, CI = 5.95, 25.95) to report suicidal behavior compared to their peers who had strong social support. Additionally, those who exhibited high levels of anger expression had 7.3 times the odds of reporting suicidal behavior (AOR = 7.30, CI = 2.51, 21.22) compared to individuals with lower anger expression. Furthermore, adolescents who experienced adverse childhood events, such as abuse, neglect, and household dysfunction, faced a sevenfold increase in the risk of suicidal behavior (AOR = 7.11, CI = 3.56, 14.20) compared to their peers.

Moreover, adolescents with a family history of suicide attempts had a 5.66 times higher risk (AOR = 5.66, CI = 2.44, 13.10) of exhibiting suicidal behavior than those without such a background. Participants who reported poor relationships with their parents also showed increased odds (AOR = 5.59, CI = 2.59, 12.08) of developing suicidal behavior compared to those with positive parental relationships. Those suffering from depression were about five times more likely (AOR = 5.32, CI = 2.64, 10.70) to engage in suicidal behavior. Similarly, adolescents facing stressful life experiences had higher odds

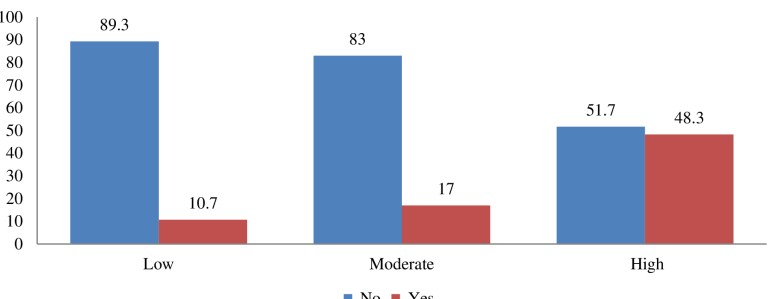

**Fig 1. Percentage distribution of suicidal behavior by level of anger expression.** (Yes): Had suicidal behavior. (No): Not had suicidal behavior.

(AOR = 2.91, CI = 1.45, 4.91) of suicidal behavior compared to their peers. Finally, participants experiencing high levels of interpersonal stress were approximately twice as likely (AOR = 2.17, 95% CI = 1.07, 4.40) to report suicidal behavior compared to those with lower levels of interpersonal stress (Table 3).

## Discussion

The primary objective of this study was to evaluate the prevalence and factors influencing suicidal behavior among adolescents residing in Bahir Dar city. The finding indicated that the prevalence of suicidal behavior was 19.8% (95% CI: 16.6, 22.8). Key factors associated with suicidal behavior included a family history of suicide, exposure to stressful life events, adverse childhood experiences, poor social support, troubled parent-child relationships, depression, and high levels of anger expression. This study underscores that suicidal behavior is a significant public health concern for adolescents in Bahir Dar city.

### Prevalence of suicidal behaviors

The present finding was in line with a study of high school students in Ethiopia that found that 22.5% had suicidal ideation and 16.2% had attempted suicide [32]. Furthermore, the results were consistent with studies conducted in Ghana among high school students, which found that 18.2% had suicidal thoughts, 22.5% had suicidal plans, and 22.2% had attempted suicide [42]; in Mozambique, where 17.7% had suicidal ideation and 18.5% had attempted suicide [43]; and in Togo (16.5% had suicidal thoughts) [44].

However, the finding in this study was higher than the school-based study done in Kut City, which reported suicidal behavior to be 8.3% [45], and Tunisia (9.6% suicidal ideation and 7.3% suicidal attempt) [46]. On the contrary, the finding was lower than studies done in Liberia, where the prevalence of suicidal attempts and suicidal ideation were 33.7% and 26.8% among adolescent students, respectively [47], Lebanese suicidal ideation was 28.9% [12], and Benin (23.2% suicidal thought and 28.3% suicidal attempt) [21]. The disparity may be explained by differences in the sample size, study setting, socio-demographic, economic, and cultural characteristics of participants, as well as by the measurement tools used; the Mini International Neuropsychiatric Interview (MINI) Suicidal Scale was used in this study [33].

### Factors associated with suicidal behavior

Suicidal behavior was 12.42 times more likely to occur in adolescents with poor social support than in those with good social support. Likewise, other studies conducted in Ethiopia showed a relationship between suicidal behavior and a lack of social support [2,32,48]. Likewise, adolescents who did not have a good relationship with their parents were 5.59 times more highly at risk for suicidal behavior than those who had a good relationship with their parents. These findings were supported by prior research conducted in China and Japan [16,49]. These relationships may arise from the fact that being

**Table 3. Bi-variable and multivariable logistic regression of suicidal behavior and associated factors among participants.**

| Variables | Category | Suicidal behavior | | COR(95%CI) | AOR(95%CI) | p-value |
|---|---|---|---|---|---|---|
| | | N (%) no | N (%) yes | | | |
| Sex | Male | 247 | 36 | 1 | | |
| | Female | 259 | 89 | 2.36(1.54,3.51) | 1.52(0.75,3.07) | 0.24 |
| Age | Early adolescent (10–13) | 64 | 11 | 1 | | |
| | Middle adolescent(14–17) | 319 | 54 | 0.98(0.49,1.99) | 0.69(0.23,2.04) | 0.50 |
| | Late adolescent(18–19) | 123 | 60 | 2.83(1.39,5.77) | 1.25(0.40,3.90) | 0.69 |
| Living status | With family | 441 | 87 | 1 | | |
| | Alone | 39 | 23 | 2.91(1.70, 5.26) | 0.47(0.17,1.27) | 0.14 |
| | Other | 26 | 15 | 2.92(1.49,5.75) | 0.98(0.31,3.05) | 0.90 |
| Family history of suicidal attempt | No | 463 | 83 | 1 | | |
| | Yes | 43 | 42 | 5.45(3.35,8.85) | **5.66(2.44,13.10)** | **≤0.01**\*\* |
| Stressful life events experience | No | 343 | 30 | 1 | | |
| | Yes | 163 | 95 | 6.66(4.25,10.45) | **2.91(1.43,5.90)** | **≤0.01**\*\* |
| Adverse child-hood experience | No | 383 | 19 | 1 | | |
| | Yes | 123 | 106 | 17.37(10.24,29.48) | **7.11(3.56,14.20)** | **≤0.01**\*\* |
| Relationship with parent | Good to very good | 465 | 77 | 1 | | |
| | Disturbed or very disturbed | 41 | 48 | 7.07(4.36,11.44) | **5.59(2.59,12.08)** | **≤0.01**\*\* |
| Grow up with biological parent | Yes | 443 | 60 | 1 | | |
| | No | 63 | 65 | 7.62(4.91,11.82) | 1.41(0.61,3.21) | 0.41 |
| Anger out expression | Low anger expression | 134 | 16 | 1 | | |
| | Moderate anger expression | 327 | 67 | 1.72(0.96,3.06) | 1.77(0.74,4.26) | 0.19 |
| | High anger expression | 45 | 42 | 7.82(4.01,15.23) | **7.30(2.51,21.22)** | **≤0.01**\*\* |
| Social support | Good social support | 364 | 16 | 1 | | |
| | poor social support | 142 | 109 | 17.46(9.98,30.55) | **12.42(5.95,25.95)** | **≤0.01**\*\* |
| Depression | No | 443 | 67 | 1 | | |
| | Yes | 63 | 58 | 6.09(3.92,9.44) | **5.32(2.64,10.7)** | **≤0.01**\*\* |
| Ever use of alcohol | No | 476 | 97 | 1 | | |
| | Yes | 30 | 28 | 4.59(2.62,8.01) | 1.50(0.53,4.20) | 0.44 |
| Current use of tobacco | No | 494 | 114 | 1 | | |
| | Yes | 12 | 11 | 3.97(1.71,9.23) | 0.26(0.59,1.14) | 0.75 |
| Interpersonal stress | Low stress high stress | 273 233 | 33 92 | 3.26(2.11, 5.04) 1 | **2.17(1.07.4.40)** | **0.03** |

*P value is significant at P<0.05, ** P value is significant at P<0.01, P value of the Hosmer and Lemeshow Test=0.939.

neglected by family and friends and not getting instrumental, informational, and emotional support from important others greatly increases feelings of worthlessness and hopelessness, which in turn contribute to suicidal behavior [50].

Adolescents who experienced negative childhood experiences (abuse, neglect, and household dysfunction) were seven times more likely to exhibit suicidal behavior than their peers. Previous research has revealed that adolescents who have experienced abuse, physical harm, or violence are more likely to engage in suicidal behavior [36]. Adolescents who had to go through stressful events in their lives had a 2.91 times higher risk of suicidal behavior than their counterparts. This finding was supported by previous studies in which suicidal behavior was significantly favored by stressful life events [46,51]. Adverse childhood experiences and stressful life events can have long-term psychological consequences,

increasing vulnerability to mental health disorders, including depression and anxiety, or intensifying pre-existing conditions, increasing suicide ideation and behavior [52–54].

Adolescents who had to go through difficult life situations had almost a threefold higher risk of suicidal behavior than their counterparts. Previous research has found that stressful life situations are much more likely to promote suicidal conduct [46,51]. Adverse childhood experiences and stressful life events can have long-term psychological consequences, including greater vulnerability to mental health concerns such as depression and anxiety, or they can trigger or worsen existing mental health difficulties, leading to an increase in suicidal thoughts and actions.

Adolescents having a family history of suicidal behavior were 5.66 times more likely to attempt suicide than those without such a history. This finding was supported by a study conducted in Fitche Town, North Shoa, Oromia region [48], while no significant relationships were discovered in Dangila town [32]. The difference could be due to the study subjects' strategies for coping, sample size, and study setting.

Participants with depression had a five times higher risk of suicidal behavior than those without depression. Previous research has indicated that depression in adolescents has a positive association with suicidal behavior [22,45,55]. Participants with high interpersonal stress had roughly twice the risk of suicidal behavior as those with low interpersonal stress. A previous study also supports that high interpersonal stress was significantly associated with suicidal behavior [56]. A potential explanation for this is that depression and interpersonal stress have often been recognized as important correlations or predictors of suicide conduct in adolescents.

Finally, the current study revealed that adolescents with higher anger expression behavior were roughly seven times more likely to report suicidal behavior than those with lower anger expression behavior. Individuals with anger-related illnesses may demonstrate a variety of cognitive and interpersonal impairments that harm their well-being and contribute to a higher risk of suicidal conduct, such as poor social problem-solving ability and reasoning. The outcomes of this study offer support to the idea that high levels of external angry emotions are significantly associated with depressed symptoms and suicidal behavior.

### Strength and limitation of the study

This study was not without limitations. First and foremost, because the study is cross-sectional, causal relationships between independent factors and suicide behaviors cannot be explained. Second, due to the topic's sensitive nature, there may have been reporting bias during data collection. Third, the study failed to account for the prevalence of other mental or psychological problems besides depression, which could have been linked to suicidal behavior. Furthermore, this study was unable to evaluate the techniques of suicide attempts. The study's strengths include the use of standard and validated tools to look at dependent and independent variables, the incorporation of characteristics concerning the COVID-19 epidemic, and being a community-based study.

### Conclusion

The prevalence of suicidal behavior was rather high in this study, indicating a substantial public health risk among teenagers in Bahir Dar that demands special attention. Suicidal behavior was significantly associated with a lack of social support, depression, a family history of suicide, not having a good relationship with their parent, adverse childhood experiences, and experiencing stressful life events. The findings suggest that mental health interventions need to be implemented within communities to enhance the relationship between parents and adolescents by raising parental awareness of adolescent mental health and integrating youth-friendly health services that promote adolescent mental well-being and offer psychosocial support. Policymakers may also consider instituting a required mental health program in schools.

### Acknowledgments

Our deepest thanks go to all study participants, data collectors, and supervisors who spent their valuable time for the good outcome of the research work.

## Author contributions

**Conceptualization:** Belete Birhan.

**Data curation:** Belete Birhan, Michael Beka, Habte Belete, Abdulbasit Sherfa, Gidey Rtbey.

**Formal analysis:** Belete Birhan, Habte Belete, Abdulbasit Sherfa, Gidey Rtbey.

**Funding acquisition:** Belete Birhan.

**Investigation:** Belete Birhan, Habte Belete.

**Methodology:** Belete Birhan, Michael Beka, Habte Belete, Abdulbasit Sherfa, Gidey Rtbey.

**Project administration:** Belete Birhan, Michael Beka, Habte Belete.

**Resources:** Belete Birhan.

**Software:** Belete Birhan, Michael Beka, Habte Belete, Abdulbasit Sherfa, Gidey Rtbey.

**Supervision:** Belete Birhan, Michael Beka, Habte Belete, Abdulbasit Sherfa.

**Validation:** Belete Birhan, Michael Beka, Habte Belete, Abdulbasit Sherfa, Gidey Rtbey.

**Visualization:** Belete Birhan, Michael Beka, Habte Belete, Abdulbasit Sherfa, Gidey Rtbey.

**Writing – original draft:** Belete Birhan, Abdulbasit Sherfa.

**Writing – review & editing:** Belete Birhan, Michael Beka, Habte Belete, Abdulbasit Sherfa, Gidey Rtbey.

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
