## [Decision Letter · Decision Letter 0]

22 Dec 2024

PONE-D-23-31943Prevalence and associated factors of suicidal behavior among adolescents at Bahir Dar city, North West Ethiopia, community based study, 2021PLOS ONE

Dear Dr. Sherfa,

Thank you for submitting your manuscript to PLOS ONE. After careful consideration, we feel that it has merit but does not fully meet PLOS ONE’s publication criteria as it currently stands. Therefore, we invite you to submit a revised version of the manuscript that addresses the points raised during the review process.**Editor Comments:**

• Make the title very clear and concise and suggest removing the year, and the title should be like: Prevalence and associated factors of suicidal behaviour among adolescents in Bahir Dar city, Northwest Ethiopia.

• This is not clear why five email IDs before the correspondent author. Please remove from your revised version all those email IDs in the title page.

Abstract section:

• Background should be succinct and justify the requirements of this research rather than long generic information, which you can talk about with contemporary statistics to back up your argument.

• The methods section should be more focused, i.e., who are your study participants (adolescents, which age group?) and what is your sampling strategy? It should be clearly stated. What validated survey tools have you used to measure the prevalence of suicidal behaviour, which is also missing in this section?

• Results section should be more succinct; particularly, you can start highlighting the prevalence and followed with the high AOR risk variables first and the lowest at last; that makes the presentation clearer.

• Conclusion should be clearly stated, which should be succinct based on your findings and provide a pragmatic recommendation to address the burden.

Introduction section

• The introduction should start with the global and country burdens and justify why this research is crucial today. You can back up your claim with current literature, but the current version highlights statistics from 2012/13, which are too old to claim the importance of this study.

• The first part of the introduction provides an overview of the global and your study country's prevalence/incident rates of suicidal behaviour, and the second section consistently highlights the risk factors of suicidal behaviours among adolescents, which should be succinct and supported by current literature.

Methods and material:

• Study design should be succinct and focused on using credible sources.

• Separate the study design and study site; the study site and participants could be put together if you want, and focus your writing very clearly.

• Please state sample size and sampling procedure in one section and make data collection in a separate section. Make data collection very clear with what the core tools you have used are and how you validated those tools, which should be succinctly stated.

• Make an "analysis" section with data analysis. State very clearly why you used (P ≤ 0.20) in the bivariate analysis instead of p < 0.05 in the multivariate logistic regression analysis.

Results section

• Make the socio-demographic characteristic section narration very succinct and also add the mean and SD age in Table 1.

• Clinical and other related factors not very clear how you measured depression and fear of COVID-19; which tools should you have used that should be clearly stated in the data collection tools section?

• Please improve your presentation style, which is not good to start academic writing in percentage, i.e., 34%, 53.6% in the quantitative presentation.

• Make very clear of the “factors associated with suicidal behaviour among adolescent participants” section, which should start presenting your findings from the highest AOR to the lowest one.

Discussion Section:

• Please organise your discussion section into two parts: 1) Prevalence of suicidal behaviours and 2) factors associated with the suicidal behaviours. Each section starts with your findings first, and that should be compared and contrasted with the global literature and provide your thoughts if particular findings are unique or already reported from literature.

Limitation section:

• Please also add the strengths of the study with limitations.

Conclusion section:

• Draw your conclusion very clearly, concisely, and coherently. What were the key outcomes of your study? Provide a pragmatic suggestion in a paragraph rather than so many subheadings that should be clear, concise, and coherent.

General comments:

• Thoroughly proofread your entire report and make the consistency of your subheadings’ capital and small letters and the presentation of AOR in all sections.

• Thoroughly review the reference list and make your reference list complete and consistent.

We look forward to receiving your revised manuscript.

Kind regards,

Sharada P Wasti, Ph.D., MSc, MHCM, MA, PGDHCM, PGCert

Academic Editor

PLOS ONE

Journal Requirements:

2. In the online submission form, you indicated that your data is available only on request from a third party. Please note that your Data Availability Statement is currently missing the name of the third party contact or institution / contact details for the third party, such as an email address or a link to where data requests can be made. Please update your statement with the missing information.

**Reviewers' comments:**

Reviewer's Responses to Questions

**Comments to the Author**

1. Is the manuscript technically sound, and do the data support the conclusions?

Reviewer #1: Yes

Reviewer #2: Yes

2. Has the statistical analysis been performed appropriately and rigorously?

Reviewer #1: I Don't Know

Reviewer #2: Yes

3. Have the authors made all data underlying the findings in their manuscript fully available?

Reviewer #1: Yes

Reviewer #2: Yes

4. Is the manuscript presented in an intelligible fashion and written in standard English?

Reviewer #1: No

Reviewer #2: Yes

5. Review Comments to the Author

Reviewer #1: It is an interesting topic. I have reviewed your manuscript and raised some concerns that needs a response from you.

1. Abstract:rewrite the result section in the abstract in more summarised way. No need of interpretation in this section.

2. Methods and materials: In the exclusion criteria, participants participants couldnot communicate well were not included? What does mean by participants couldnot commnunicate well? How many of the participants found to have this problem? I think justify it or avoid it. Additionally, how many of the participants found to have severe illnesses and excluded from the study?

Data quality control: you have missed it in your manuscript, write how you assure to the data quality?

Reviewer #2: COMMENTS OF THE MANUSCRIPT:

Prevalence and associated factors of suicidal behavior among

adolescents at Bahir Dar city, North West Ethiopia, community based study, 2021

ABSTRACT

Under methods; The sentence “To determine the independent components related with suicidal behaviour, bi-variable and multivariable logistic regression were use” should read have “d” added to use - to read: "To determine the independent components related with suicidal behaviour, bi-variable and multivariable logistic regression were used.

INTRODUCTION

Line 4: In the last sentence, change “it” to It (I to be in upper case).

Do editing review of the write - especially with punctuation, staring letters,

Globally there have been an estimated 804 000 suicide deaths arose in 2012, representing a twelve-monthly global age-standardized suicide rate of 11.4 per 100 000 population (15.0 for males and 8.0 for females) - reframe this sentence and PROVIDE MORE RECENT DATA, 2012 as a standalone data is obsolete, except authors can show a trend. IF SO, CITE THE SOURCE OF THE INFORMATION!

Sub-Saharan Africa there are over 34,000 (IQR 13,141 to 63,757) suicides per year, with an overall incidence rate of 3.2 per 100,000 populations (13) --- THIS INFORMATION IS OBSOLTE FOR A RECENT FACT (10 years ago). UPDATE THE INFORMATION

Citations - (13),(14) and (15) are outdated - provide more recent data.

Comments on Introduction: UPDATE THE LITERATURE

METHODS AND MATERIALS

Under Participants: Re frame the sentence. Correct the grammar and use formal language and avoid use of words like “couldn’t.”

Under Measurements - EDIT THE SECTION - Grammar

Under Ethical considerations - Ethical clearance was obtained from Institutional review board (IRB) of Bahir Dar University, College of Medicine and Health Science - INSERT THE APPROVAL NUMBER/DATE or make reference to it as an appendix,

Line 3 (last part) - These sentences: “The aims of the study were explain for the study participants and data were collected after assent and written consent. Adolescents aged 18 years and above were gave informed written consent to participate. Participants under the age of 18 were verbal assent to the study and then written consent was obtained from their parents /guardians/ on their behalf” NEED TO SPECIFICALLY CORRECTED.

RESULTS

Line 3 under Psychosocial and Substance Related Factors: …...14.1% (89) of respondents don't have a good relationship with…..USE FORMAL LANGUAGE

Under “Factors Associated with suicidal behavior among Adolescent participant”, review the first sentence structure - variables cannot be candidates

Comments on other sections: DO THOROUGH EDITING: LANGUAGE AND STRUCTURE

6. PLOS authors have the option to publish the peer review history of their article (what does this mean? ). If published, this will include your full peer review and any attached files.

**Do you want your identity to be public for this peer review?** For information about this choice, including consent withdrawal, please see our Privacy Policy .

Reviewer #1: **Yes: ** Agmas Wassie Abate

Reviewer #2: **Yes: ** Dr Emmanuel Ejembi Anyebe

---

## [Author Response · Author response to Decision Letter 1]

28 Jan 2025

To: Sharada P Wasti, Ph.D., MSc, MHCM, MA, PGDHCM, PGCert

Academic Editor

PLOS ONE

Thank you for considering our manuscript and for arranging for it to be reviewed by two reviewers. We have tried to address your comments and the comments / suggestions from the two reviewers. In the Rebuttal letter, we copy each of the comments in bold and provide the RESPONSE underneath. We provide a clean manuscript file version of the revised manuscript with line numbering and this is uploaded as a separate file labeled 'Clean manuscript final version'. We also provide an edited manuscript with track changes and this is uploaded as a separate file labeled as ‘Edited manuscript with Track changes’. We hope we have satisfactorily addressed all the comments and hope that our paper may now be suitable for publication in your journal.

Best wishes

Abdulbasit Sherfa

Corresponding author

Editor comment

Make the title very clear and concise and suggest removing the year, and the title should be like: Prevalence and associated factors of suicidal behavior among adolescents in Bahir Dar city, Northwest Ethiopia.

Response: Thank you for your suggestion and we corrected the title of the article in the revised manuscript as follow:

Prevalence and associated factors of suicidal behavior among adolescents in Bahir Dar city, Northwest Ethiopia

• This is not clear why five email IDs before the correspondent author. Please remove from your revised version all those email IDs in the title page.

Response: we removed in the revised manuscript

Abstract section

• Background should be succinct and justify the requirements of this research rather than long generic information, which you can talk about with contemporary statistics to back up your argument.

Response: Suicide is a significant public health issue worldwide and the third leading cause of death among 15–19-year-olds. In Ethiopia, while suicidal behaviors among adolescents have a considerable impact, there is limited information on these behaviors, especially in community settings. Therefore, this study aimed to assess the prevalence and associated factors of suicidal behavior among adolescents in Bahir Dar city

• The methods section should be more focused, i.e., who are your study participants (adolescents, which age group?) and what is your sampling strategy? It should be clearly stated. What validated survey tools have you used to measure the prevalence of suicidal behaviour, which is also missing in this section?

Response: A community-based cross-sectional study was conducted from May 1 to May 30, 2021, involving 692 adolescents aged 11 to 19 years in Bahir Dar city, Northwest Ethiopia. Mini-International Neuropsychiatric Interview (MINI) was used to assess suicidal behaviors. Participants were selected using a multistage sampling method. Data collection was performed through face-to-face interviews employing semi-structured and standardized questionnaires. For processing and analysis, the acquired data was entered into Epi-Data version 3.1 and exported to Statistical Package for Social Science version 24. To assess the strength of associations between the outcome and predictor variables, a logistic regression analysis model with adjusted odds ratios was applied. A p-value of less than 0.05 was considered statistically significant.

• Results section should be more succinct; particularly, you can start highlighting the prevalence and followed with the high AOR risk variables first and the lowest at last; that makes the presentation clearer.

Response: The prevalence of suicidal behavior was found to be 19.8% with 95% CI (16.6, 22.8).

Regarding predictive variables: low social support [AOR = 12.42, CI = 5.95, 25.95], Expressed more anger [AOR = 7.30, CI = 2.51, 21.22], Negative childhood experiences [AOR = 7.11, CI = 3.56, 14.20], Family history of suicidal attempts [AOR = 5.66, CI = 2.44, 13.10], Have no a good relationship with their parents [AOR = 5.59, CI = 2.59, 12.08], Depression [AOR = 5.32, CI = 2.64, 10.70], Stressful life events [AOR = 2.91, CI = 1.45, 4.91], and Interpersonal stress [AOR = 2.17, 95% CI 1.07, 4.40] were significantly increased odds of suicidal behavior among participants.

• Conclusion should be clearly stated, which should be succinct based on your findings and provide a pragmatic recommendation to address the burden.

Response: Conclusion and recommendation: The prevalence of suicidal behavior among participants was high in this study, influenced by factors such as low social support, high anger expression, negative childhood experience, family history of suicidal attempts, poor parental relationships, stressful life events, and depression. Therefore, it is essential to enhance social support systems, provide mental health resources, and implement community-based interventions focused on improving family dynamics and reducing childhood adversity.

Introduction section

• The introduction should start with the global and country burdens and justify why this research is crucial today. You can back up your claim with current literature, but the current version highlights statistics from 2012/13, which are too old to claim the importance of this study.

• The first part of the introduction provides an overview of the global and your study country's prevalence/incident rates of suicidal behaviour, and the second section consistently highlights the risk factors of suicidal behaviours among adolescents, which should be succinct and supported by current literature.

Response: Thank you for your comments and we tried to update our introduction based on current literatures as follow:

Suicide is death caused by an act of self-harm that is intended to be lethal. Suicidal behavior include completed suicide, attempted suicide, and suicidal ideation, which includes suicide-related thoughts, plans, and acts [1]. Suicidal behaviors is a serious global public health issue, accounting for more than 2.4% of the global disease burden by 2020[2]. An estimated 703,000 people died by suicide worldwide in 2019, with low- and middle-income countries (LMICs) accounting for roughly three-fourths of all deaths. Suicide was the 4th and the 3rd leading cause of death for those aged 15 to 29 and 15 to 19, respectively, and nearly 9 out of 10 were from LMICs, which are home to nearly 90% of all adolescents worldwide [3-6]. Africa has six of the ten highest suicide rates in the world, with 11 suicide fatalities per 100,000 people annually [7]. According to WHO (2021), about 34,000 suicides are reported in Africa each year, accounting for approximately 3.2 per 100,000 people [8, 9]. According to pooled data, the prevalence of suicidal ideation and attempted suicide in Ethiopia ranged from 1% to 55%, and 0.6% to 14% [10].

Adolescence is a critical period marked by physical, emotional, and societal changes, in which people might engage in a variety of risky behaviors, including suicidal ideation, suicide attempts, and suicide [11-13]. Suicide is a principal cause of death among adolescents. Self-harm is the most risk factor for suicide, yet the majority of self-harm does not come to the attention of health services [14]. Suicidal behavior, deliberate self-harm, and non-suicidal self-injury are significant precursors of suicide in children and adolescents [15]. Adolescent psychiatric emergencies are a leading cause, with suicidal attempts and deaths being the strongest predictors [16]. Suicidal thoughts and suicidal behaviors develop during adolescence and peak late in adolescence and early adulthood [17].

Worldwide, the possible risk factors of suicide include victimization (bullying, sexual harassment), poor psychological state related to depression, phobic disorders, anxiety, alcohol use disorder, child abuse, impulsivity, and lack of parental understanding [12, 18, 19]. Studies show that mental state issues, substance use, genetic and biological factors, poor physical health, physical disability, and family-environmental factors increase the risk of suicidal behaviors in adolescents [16, 20, 21]. Suicide is a prevalent issue among adolescents, with those who have experienced suicidal ideation and attempts being significantly more likely to commit suicide [22].

Suicide costs billions of dollars in lost productivity and healthcare expenditures, requiring comprehensive prevention initiatives, enhanced mental health services, public awareness campaigns, and community engagement [23]. Suicide prevention requires robust healthcare systems, universal health coverage, and community engagement. WHO's tool kit aims to address stigma, improve knowledge, and provide social support [24]. In 2013, the 66th World Health Assembly adopted the WHO's Mental Health Action Plan, focusing on suicide prevention to reduce suicide rates by 10% by 2020[25].

Adolescent suicidal behavior is a neglected public health issue, especially in LMICs including Ethiopia [26]. Underinvestment on mental health and lack of national health insurance coverage in LMICs make it more difficult to get resources and help, particularly for young people between the ages of 15 and 29, who are faced with trauma, social stigma, and financial difficulties [27]. It is a multipart and multidimensional phenomenon stemming from the interaction of several factors [28]

Suicide rates are increasing rapidly among young individuals, particularly males aged 15-24, and the rate is still growing [10]. Even though LMICs account for the bulk of suicides, high-income Western nations are the source of most of the knowledge about suicidal behaviors. Because there aren't many studies on teenagers in Ethiopia, little is known regarding the prevalence and determinants of suicide behavior (suicidal thoughts, suicidal plans, and suicidal attempts). Therefore, this study aimed to evaluate the prevalence of suicidal behavior and its contributing factors among adolescents.

Methods and material:

• Study design should be succinct and focused on using credible sources.

• Separate the study design and study site; the study site and participants could be put together if you want, and focus your writing very clearly.

• Please state sample size and sampling procedure in one section and make data collection in a separate section. Make data collection very clear with what the core tools you have used are and how you validated those tools, which should be succinctly stated.

Response: we corrected methods and material section as follow:

Study setting

The study was conducted in Bahir Dar city Northwest Ethiopia which is 565 kilo meters far from Addis Ababa. According to Bahir Dar city mayor office report in 2019/2020, the estimated population in the city was 373,073. From the total population the number of adolescent (between 10-19 years) was 84,382. The city has one comprehensive specialized hospital, one primary hospital, one specialized public hospital and four private hospitals, which give service to the city and surrounding population. However, only the comprehensive specialized and specialized hospitals have inpatient psychiatric service [29, 30].

Study design and period

A community-based cross-sectional study was conducted in Bahir Dar city Northwest Ethiopia from May1-30, 2021.

Study participants

All adolescents (aged 10-19 years old) residing in Bahir Dar city were source population whereas adolescents who had been living for the last six months within the selected sub cities of the town were study population. Moreover, households were taken as sampling unit and selected eligible adolescent was considered as study unit. Adolescents who were residing for last six months in Bahir Dar city selected sub cities were included and adolescents from selected households who were severely ill and unable to respond to the interviewer properly were excluded.

Sample size determination and sampling methods

Sample size was determined using a single population proportion formula using the proportion of suicidal behavior 22.5% from previous study in Ethiopia [31] and 95% confidence interval was used, 4% margin of error, design effect 1.5 and by assuming a 10% non-response rate, the final sample size was 692.

Multistage sampling technique was used to select the sub-cities (three) from the total of six and to select respective administrative kebeles (the smallest administrative unit) (six) from total of seventeen. The households in the administrative kebeles were selected by systematic random sampling technique after identifying an initial starting household by use of random number. Eligible adolescents in the selected household were further selected and interviewed. Only one adolescent member of the house hold was selected by lottery method for the interview towards suicide behavior

Data collection

The questionnaire included items to assess socio-demographic information, suicidal behavior, self-esteem, social support, adverse childhood experience, anger expression, stressful life events, clinical factors, behavioral factors and interpersonal stress. Three health extension workers collected the data using face-to-face interviews. The data collection was supervised by a psychiatry nurse (bachelor’s degree holder). Training on the questionable and data collection proceed was provided for the supervisor and data collectors.

Measurements

Suicidal behaviors: Participants who answered "yes" to at least one of the six questions in the Mini-International Neuropsychiatric Interview (MINI) were categorized as displaying suicidal behavior. The MINI shows high internal consistency (Cronbach’s alpha = 0.84) [32].

Social support: The participants' social support was assessed using the Oslo-3 social support scale (25). Based on their scores, participants were classified as follows: those with scores between "3–8" were considered to have poor social support, scores of "9–11" indicated moderate support, and scores ranging from "12-14" represented strong social support.

Depression: Depression was assessed by a nine-item Patient Health Questionnaire-9, which has four response categories referring to the amount of time that the symptom was present (not at all (0), several days (1), more than half of the days (2), nearly every day (3)), with a total score ranging from 0 to 27 [33]. Adolescents who scored ≥5 in PHQ-9 were considered as having depression [34]

Adverse child hood experience: The ACE Questionnaire (Adverse Childhood Experiences Questionnaire) was used to assess the impact of traumatic childhood experiences, including abuse, neglect, and household dysfunction. It consists of questions about various adverse experiences [35].

Self-esteem: The participant's self-esteem was assessed using the Rosenberg self-esteem scale. People scoring between 15 and 25 are average. A score of less than 15 suggests low self-esteem and a score greater than 25 suggests high self-esteem [36].

Anger Expression: The participant's expression of anger was evaluated using the Spielberger Anger-Out Expression Scale, which includes eight items designed to assess the coping strategies individuals employ, especially in relation to outward expressions of anger. This scale categorizes anger expression into three levels: low anger expression (scores below 10), moderate anger expression, and high anger expression (scores of 15 or above) [37].

Stress full life events: According to the SLESQ definition, individuals who reported a positive response to at least one of the eleven specific categories or the two general categories of events—such as enduring a life-threatening accident, experiencing physical or sexual abuse, or witnessing a murder or assault—were classified as having undergone stressful life events [38].

Interpersonal stress: Interpersonal stress was asses by using Bergen Social Relationships Scale BSRS. The scale consists of six items. Individuals with high scores (above the mean) of BSRS have high interpersonal stress [39].

Fear of corona virus. The assessment of fear associated with the coronavirus was carried out using a specially designed seven-item scale. Participants indicated their level of agreement with a range of statements using a five-point Likert scale, which included the options "strongly disagree," "disagree," "neutral," "agree," and "strongly agree". Individuals with high scores (above the mean) hav

---

## [Editor Report · Decision Letter 1]

4 Feb 2025

PONE-D-23-31943R1Prevalence and associated factors of suicidal behavior among adolescents in Bahir Dar city, Northwest EthiopiaPLOS ONE

Dear Dr. Sherfa,

Thank you for submitting your manuscript to PLOS ONE. After careful consideration, we feel that it has merit but does not fully meet PLOS ONE’s publication criteria as it currently stands. Therefore, we invite you to submit a revised version of the manuscript that addresses the points raised during the review process.

We look forward to receiving your revised manuscript.

Kind regards,

Sharada P Wasti, PhD

Academic Editor

PLOS ONE

Journal Requirements:

Additional Editor Comments:

I greatly appreciate all authors for your thorough review and making revisions to the manuscript, which has vastly improved its clarity. However, there are still some remaining concerns and areas for improvement in this manuscript. I kindly request that you carefully review and address each of the concerns indicated below before reaching a final decision:

Abstract section:

• Background—remove “therefore” from your last sentence.

• Methods: remove the date of “May 1-May 30” and keep only “May 2021”.

• Talk first about the study design, followed by the study sites, respondents, sample size and sampling technique, data collection, and analysis, which should be coherent rather than random. Your second sentence, “Mini International Neuropsychiatric Interview (MINI) was used to assess suicidal behaviours,” is not clear about what you intend to state in the current form.

• Results: Correct your way of presenting of inside small bracket (AOR XX, 95%CI xx to xx) which should be like (AOR 2.17, 95%CI 1.07 to 4.40) and make it consistent in the entire report.

• Remove “recommendation” from your conclusion and recommendation; keep only the conclusion in the section heading and follow at the end of your conclusion section as well.

• Recommendation sentences should be pragmatic to combat the burden.

Introduction section:

• Define first-time use abbreviation, i.e. WHO

Methods and material section:

• The study design section should be clearly articulated using credible sources to back up your claims.

• Under the study setting, explain the distinction between “comprehensive specialised and specialised hospitals” in Ethiopia.

• Sample size determination and sampling methods section: delete the word “determination” and make the section title "Sample size and sampling methods.". If you keep the sample size calculation formula, which you did in your previous version, that would be perfect to explain how 692 was calculated.

• Add methods and tools under the “data collection” section, and the section title should be "Data collection methods and tools.".

• Remove “measurement” and add tools, and your section title looks like “Data collection tools.”

• Remove the separate sub-section of “data quality control” and use key quality control measures information in the above Data Collection Methods and Tools section.

• Remove “processing” and keep only Data analysis.

• What was the reason for including a bivariate analysis of a p-value of less than 0.2 in multivariable logistic regression analysis instead of a p-value of <0.05? Explain with credible sources to back up your argument.

Results

• Proofread and state “Results”

• Make sure the small and capital letters of section sub-headings are constant across the report, such as “Socio-demographic Characteristics of the Respondents,” which are inconsistent across the section.

• In a quantitative paper, the best way is to use respondents instead of participants and maintain consistency.

• Check the entire writing and thoroughly proofread, i.e., after the full stop in Table 1, see Tables 2 and 3, which are inconsistent and require careful proofreading.

• Refer to Figures 1 and 2 individually rather than in the same location. So, describe the first results and signpost that figure, then put the second figure's content and figure number on the signpost.

• Correct the Y-axis of both of your figures, i.e., is this a percentage or the number of responders, which should be a percentage based on your data? I would recommend keeping only Figure 2 in the body of content and removing Figure 1.

• Keep only “Factors Associated with Suicidal Behaviour among Adolescents” and remove “participant” from the section sub-section heading. Remove the first paragraph's information, which should be clearly articulated in your data analysis section, not under the findings section.

• The overall findings section should be succinct and clearly describe the findings without providing any methodological details.

Discussion section:

• Outline the first paragraph of what you are going to discuss in this section, which I gave you last suggestions but did not address yet. Your discussion should outline two sections, i.e., the prevalence of suicidal behaviour and factors associated with suicidal behaviour, and critically articulate as per the last provided suggestions.

• Carefully review your strengths and limitations section and make it very clear and correct statements instead of very generic statements.

Conclusion section:

• Remove “and recommendation” and keep only the conclusion of the section title.

• Provide pragmatic recommendations to address the burden instead of generic and vague statements.

Proofreading is required to enhance writing patterns and ensure consistency, and this should be reflected in your track change version. Several sections i.e. inconsistent small and capital letters, table/figure notations and using notations after a full stop.

---

## [Author Response · Author response to Decision Letter 2]

12 Mar 2025

To: Sharada P Wasti, Ph.D

Academic Editor

PLOS ONE

Thank you for considering our manuscript and for arranging for it to be reviewed by two reviewers. We have tried to address your comments and the comments / suggestions from the two reviewers. In the Rebuttal letter, we copy each of the comments in bold and provide the RESPONSE underneath. We provide a clean manuscript file version of the revised manuscript with line numbering and this is uploaded as a separate file labeled 'Clean manuscript final version'. We also provide an edited manuscript with track changes and this is uploaded as a separate file labeled as ‘Edited manuscript with Track changes’. We hope we have satisfactorily addressed all the comments and hope that our paper may now be suitable for publication in your journal.

Best wishes

Abdulbasit Sherfa

Corresponding author

Journal Requirements:

Response: Thank you for your recommendations; we evaluated the reference list and attempted to complete and correct some references, such as reference numbers 5–8, 11, 24, 25, and 27 in the revised article. In addition, four references (Refs. 31, 49, 52, 53, and 54) have been added to the amended manuscript’s methods and discussion section.

Additional Editor Comments:

I greatly appreciate all authors for your thorough review and making revisions to the manuscript, which has vastly improved its clarity. However, there are still some remaining concerns and areas for improvement in this manuscript. I kindly request that you carefully review and address each of the concerns indicated below before reaching a final decision

Abstract section:

Background—remove “therefore” from your last sentence.

Response: Thank you for your comments and we removed it in the revised manuscript as follow:

Suicide is a significant public health issue worldwide and the third leading cause of death among 15–19-year-olds. In Ethiopia, while suicidal behaviors among adolescents have a considerable impact, there is limited information on these behaviors, especially in community settings. The study aimed to assess the prevalence and associated factors of suicidal behavior among adolescents in Bahir Dar city.

Methods: remove the date of “May 1-May 30” and keep only “May 2021”.

• Talk first about the study design, followed by the study sites, respondents, sample size and sampling technique, data collection, and analysis, which should be coherent rather than random. Your second sentence, “Mini International Neuropsychiatric Interview (MINI) was used to assess suicidal behaviors,” is not clear about what you intend to state in the current form.

Response: Thank you for your suggestion and comments and we tried to correct in the revised manuscript as follow:

Methods: A community-based cross-sectional study was conducted in May 2021 in Bahir Dar City, Northwest Ethiopia, with 692 adolescents aged 11 to 19. Participants were chosen using a multistage sampling procedure. Face-to-face interviews were used to collect data, along with semi-structured and standardized questionnaires. Participants were considered suicidal if they replied "yes" to at least one of the six Mini International Neuropsychiatric Interview questions. The data were processed using Epi-Data version 3.1 and analyzed with the Statistical Package for Social Science version 24. Logistic regression analysis with adjusted odds ratio was used to determine the relationship between outcome and predictor variables. A p-value of <0.05 was considered statistically significant.

• Results: Correct your way of presenting of inside small bracket (AOR XX, 95%CI xx to xx) which should be like (AOR 2.17, 95%CI 1.07 to 4.40) and make it consistent in the entire report.

Response: Thank you for your suggestion. We have made the necessary corrections in the revised manuscript as per your request, using small brackets. However, upon reviewing several papers published in PLOS ONE, we noticed that the range of confidence intervals is often separated by a hyphen or comma rather than the word "to." To ensure consistency throughout the revised manuscript, we have opted to use commas and we corrected it as follow:

Results: The prevalence of suicidal behavior was found to be 19.8% with 95% CI (16.6, 22.8). Regarding predictive variables: low social support (AOR = 12.42, CI = 5.95, 25.95), expressed more anger (AOR = 7.30, CI = 2.51, 21.22), negative childhood experiences (AOR = 7.11, CI = 3.56, 14.20), family history of suicidal attempts (AOR = 5.66, CI = 2.44, 13.10), have no a good relationship with parents (AOR = 5.59, CI = 2.59, 12.08), depression (AOR = 5.32, CI = 2.64, 10.70), stressful life events (AOR = 2.91, CI = 1.45, 4.91), and interpersonal stress (AOR = 2.17, 95% CI 1.07, 4.40) were significantly increased odds of suicidal behavior among respondents.

• Remove “recommendation” from your conclusion and recommendation; keep only the conclusion in the section heading and follow at the end of your conclusion section as well.

• Recommendation sentences should be pragmatic to combat the burden

Response: Thank you for your suggestion and we tried to correct it in the revised manuscript and also we tried to make our recommendation pragmatic to combat the burden of suicidal behavior among adolescents as follows:

Conclusion: The prevalence of suicidal behavior was higher in this study area. Depression, stressful life events, poor parental relationships, negative childhood experiences, high anger expression, low social support, and a family history of suicidal attempts were the significant contributors. As a result, mental health interventions should focus on providing psychosocial support to adolescents, improving the relationship between adolescents and their parents by raising parents' awareness of adolescent mental health, and incorporating youth-friendly health services that promote mental well-being. Policymakers can also consider making mental health education mandatory in schools.

Introduction section:

• Define first-time use abbreviation, i.e. WHO

Response: We have corrected it in the revised manuscript and defined it as 'World Health Organization.'"

Methods and material section:

• The study design section should be clearly articulated using credible sources to back up your claims.

Response: Thank you for your feedback; we attempted to look into the study design section and modified it in the revised version, as follows:

In May 2021, a community-based cross-sectional survey was carried out in Bahir Dar City, Northwest Ethiopia. The method was used to assess the prevalence of suicide behaviors and associated factors in the population at a specific point in time. Cross-sectional studies are frequently employed in public health research to assess the prevalence of health problems and their factors (1).

• Under the study setting, explain the distinction between “comprehensive specialized and specialized hospitals” in Ethiopia.

Response: We have explained the distinction between “comprehensive specialized and specialized hospitals” in Ethiopia in the revised manuscript on the study setting part as follows:

There are three public hospitals in the city: one primary, one specialized, and one comprehensive. While specialized hospitals concentrate on a single area of healthcare and treat cases within their field, comprehensive specialized hospitals in Ethiopia offer a wider variety of advanced services across many specializations and handle more difficult and diversified cases. In addition, the city has four private hospitals. Inpatient psychiatric cares are only offered by comprehensive specialized and specialized public institutions

• Sample size determination and sampling methods section: delete the word “determination” and make the section title "Sample size and sampling methods.". If you keep the sample size calculation formula, which you did in your previous version, that would be perfect to explain how 692 was calculated.

Response: Thank you for your suggestion. We removed the word “determination” and we made the section title "Sample size and sampling methods” in the revised manuscript as follows:

Sample size and sampling methods

The sample size was determined with a single population proportion formula, based on a previous reported suicide behavior prevalence of 22.5% from a study done in Ethiopia (2). A 95% confidence interval, a 4% margin of error, and a design effect of 1.5 were employed.

Sample size was calculated as: N=Zα/2 × P (1-P)/W2.

Where Zα/2 = confidence level (1.96) at CI of 95%

N= Sample size

P=22.5% (proportion of adolescents who have suicidal ideation)

W= margin of sampling error.

1- P= sample error proportion of adolescents who have no suicidal ideation

N = (1.96)² (0.225) (1-0.225) / (0.04)2 = 419, then multiply by 1.5 for design effect, which gives 629. By assuming a 10% non-response rate, the final sample size was 692.

The multistage sampling method was employed to choose three sub-cities from a total of six and to choose six corresponding administrative kebeles (the smallest administrative entity) from a total of seventeen. The households in the administrative kebeles were selected by systematic random sampling technique after identifying an initial starting household by use of a random number. Eligible adolescents in the selected household were further selected and interviewed. Only one adolescent member of the household was selected by the lottery method for the interview on suicide behavior.

• Add methods and tools under the “data collection” section, and the section title should be "Data collection methods and tools."

Response: We have made necessary corrections to the title and also included the data quality control section in data collection methods and tools in the revised manuscript as follows:

Data collection methods and tools

The questionnaire included items to assess socio-demographic information, suicidal behavior, self-esteem, social support, adverse childhood experience, anger expression, stressful life events, clinical factors, behavioral factors, and interpersonal stress.

To ensure the data quality, the questionnaire was first developed in English, translated into Amharic language, and translated back into English by different experts to check its consistency. The data was collected using an interviewer-administered questionnaire from the selected household by three health extension workers, one BSc psychiatry nurse supervisor, and the principal investigator engaging in the supervision.

The data collectors and the supervisor were given one-day training by the principal investigator. A pre-test was carried out on 5% of the respondents (35 adolescents) in Woreta town, and according to the pre-test, there was no need for a modification for the questionnaire. The supervisor and principal investigator were closely following the day-to-day data collection process for completeness, clarity, and consistency on a daily basis.

• Remove “measurement” and add tools, and your section title looks like “Data collection tools.”

Response: We replaced “measurement” by “Data collection tools”

• Remove the separate sub-section of “data quality control” and use key quality control measures information in the above Data Collection Methods and Tools section.

Response: we removed the separate sub-section of “data quality control” and put under data collection method section in the revised manuscript as follow:

The questionnaire included items to assess socio-demographic information, suicidal behavior, self-esteem, social support, adverse childhood experience, anger expression, stressful life events, clinical factors, behavioral factors, and interpersonal stress.

To ensure the data quality, the questionnaire was first developed in English, translated into Amharic language, and translated back into English by different experts to check its consistency. The data was collected using an interviewer-administered questionnaire from the selected household by three health extension workers, one BSc psychiatry nurse supervisor, and the principal investigator engaging in the supervision.

The data collectors and the supervisor were given one-day training by the principal investigator. A pre-test was carried out on 5% of the respondents (35 adolescents) in Woreta town, and according to the pre-test, there was no need for a modification for the questionnaire. The supervisor and principal investigator were closely following the day-to-day data collection process for completeness, clarity, and consistency on a daily basis.

• Remove “processing” and keep only Data analysis.

Response: we removed “processing” and made it as per your request “Data analysis”

In the revised manuscript as follow:

Data analysis

The data was entered into Epi data version 3.1 and exported to SPSS version 24 for analysis. Necessary data processing like recoding, categorizing, merging, computing, and counting was done before the actual data analysis. After data processing, descriptive statistics like measures of central tendency (mean, median, and mode) and measures of dispersion were used for continuous variables and frequency count and proportion were used to summarize categorical variables. All variables with a p-value of less than 0.2 in the bivariate logistic regression analysis were entered into the multivariable logistic regression model to identify factors associated with suicidal behavior. The adjusted odds ratios (AORs) with 95% confidence intervals were used to assess the strength of associations between the outcome and predictor variables. The p-value of <0.05 was considered significant.

• What was the reason for including a bivariate analysis of a p-value of less than 0.2 in multivariable logistic regression analysis instead of a p-value of <0.05? Explain with credible sources to back up your argument.

Response: We valued this inquiry and endeavored to elucidate the rationale for incorporating a bivariate analysis with a p-value of less than 0.2 in multivariable logistic regression analysis, rather than a p-value of less than 0.05, as follows:

Employing a p-value threshold of <0.2 in bivariate analysis, as opposed to a p-value of <0.05 in multivariable logistic regression, is grounded on statistical and practical considerations that facilitate the formation of a robust explanatory model.

Initially, employing a strict threshold of p-value < 0.05 in the bivariate analysis may lead to the exclusion of variables that are not statistically significant independently but could attain significance or relevance when adjusted for other variables in the multivariable model; thus, variables with p-values ranging from 0.05 to 0.2 may still be pertinent factors. Secondly, a variable might act as a confounder, so if we use a p-value < 0.05, it might result in the exclusion of such confounding variables, which in turn leads to biased estimates in the multivariable model. Thirdly, in many studies, especially exploratory or observational research, the goal is to identify all possible predictors of the outcome, even if their individual significance is marginal (3, 4).

Additionally, employing a p-value of <0.2 instead of the stringent threshold is significant for considerations of statistical power. This means in smaller datasets like ours, the ability to detect a true effect (power) is limited, so using a stricter threshold (p < 0.05) might fail to detect variables that are truly associated with the outcome but have smaller effect sizes. While all these mentioned reasons would balance errors, that means reducing the risk of Type II errors (missing important variables) at the cost of a slightly higher risk of Type I errors

---

## [Editor Report · Decision Letter 2]

21 Mar 2025

PONE-D-23-31943R2Prevalence and associated factors of suicidal behavior among adolescents in Bahir Dar City, Northwest EthiopiaPLOS ONE

Dear Dr. Sherfa,

Thank you for submitting your manuscript to PLOS ONE. After careful consideration, we feel that it has merit but does not fully meet PLOS ONE’s publication criteria as it currently stands. Therefore, we invite you to submit a revised version of the manuscript that addresses the points raised during the review process.

**ACADEMIC EDITOR:** Thank you for I greatly appreciate all authors for your thorough review and making revisions to the manuscript, which has vastly improved its clarity. However, there are still some remaining concerns and areas for improvement in this manuscript. I kindly request that you carefully review and address each of the concerns indicated below before reaching a final decision. Please ensure that your decision is justified on PLOS ONE’s publication criteria  and not, for example, on novelty or perceived impact.

We look forward to receiving your revised manuscript.

Kind regards,

Sharada P Wasti, PhD

Academic Editor

PLOS ONE

Journal Requirements:

Additional Editor Comments:

I greatly appreciate all authors for your thorough review and making revisions to the manuscript, which has vastly improved its clarity. However, there are still some remaining concerns and areas for improvement in this manuscript. I kindly request that you carefully review and address each of the concerns indicated below before reaching a final decision:

Abstract section:

• Methods section: Present clearly and concisely and also add the suicidal measurement tool instead of grading criteria. Clear and concise, this sentence: The data were processed using Epi-Data version 3.1 and analysed with the Statistical Package for Social Science version 24.

• Findings section: Look at the last feedback in your findings section. The presentation should be succinct with a small bracket (AOR XX, 95% CI XX to XX), which should be like (AOR 2.17, 95% CI 1.07 to 4.40), and make it consistent in the entire report. But it still has not been fixed in the abstract section. So make the consistency of your findings presentations, i.e., at (AOR 7.30, 95% CI 2.51 to 21.22), negative childhood experiences (AOR 7.11, 95% CI 3.56 to 14.20) to all abstract and FINDINGS sections for the consistency with neat and clean presentation.

• Conclusion section: make the section very clear and concise.

• Remove the corresponding author's phone number, i.e., Phone: +251-913-69-28-80.

•

Findings section:

• All table titles should be short and clear as per the last feedback and made consistent, i.e., Table 1:, Table 2., and Table 3.

• The title of Figure 1 should be short and clear, i.e., Figure 1: Distribution of suicidal behaviour by level of anger expression

• Table two's title should be shorter as per earlier feedback, and category variables need to be made consistent where category variables are inconsistent with other tables, i.e., the first letters are small letters, so make it consistent with other tables.

• Figure 1 removes the table and provides the % value at the top of the bar diagram and removes the below tables and keeps the legend.

The discussion section should be thoroughly reviewed and succinctly presented without repeating all findings with 95% CI values, but you can use AOR only and compare and contrast with global and national empirical evidence and provide your thoughts on society.

• After completing all revisions, please fully proofread to make it clear and coherent.

---

## [Author Response · Author response to Decision Letter 3]

29 Mar 2025

Academic Editor

PLOS ONE

Thank you for considering our manuscript. We have tried to respond to your comments and recommendations. In the response letter, we bold each of the comments and include the RESPONSE beneath. We give a clean manuscript file version of the amended manuscript with line numbering, which is submitted as a separate file named 'Manuscript.' We also include an amended manuscript with track changes, which is uploaded as a separate file titled 'Revised manuscript with track changes.' We belief we sufficiently addressed all comments and are pleased that our manuscript is now suitable for publication in your journal.

Best wishes

Abdulbasit Sherfa

Corresponding author

Journal Requirements:

Response: Thank you for your recommendations; we evaluated the reference list and attempted to complete and correct some references.

Additional Editor Comments:

I greatly appreciate all authors for your thorough review and making revisions to the manuscript, which has vastly improved its clarity. However, there are still some remaining concerns and areas for improvement in this manuscript. I kindly request that you carefully review and address each of the concerns indicated below before reaching a final decision:

Abstract section:

• Methods section: Present clearly and concisely and also add the suicidal measurement tool instead of grading criteria. Clear and concise, this sentence: The data were processed using Epi-Data version 3.1 and analysed with the Statistical Package for Social Science version 24.

Response: Thank you for your suggestion and we tried to make it more concise and clear on the revised manuscript as follows:

Methods: A community-based cross-sectional study involving 692 adolescents was conducted in Bahir Dar City, Northwest Ethiopia, in May 2021. Participants were selected through a multistage sampling technique. Semi-structured and standardized questionnaires were employed to gather data via face-to-face interviews. Suicidal behavior was evaluated using the six questions of the Mini International Neuropsychiatric Interview. Epi-Data version 3.1 was utilized for data entry, whereas Statistical Package for Social Science version 24 was employed for analysis. Logistic regression with an adjusted odds ratio determined the relationship between outcome and factor variables. A p-value of <0.05 was considered statistically significant.

• Findings section: Look at the last feedback in your findings section. The presentation should be succinct with a small bracket (AOR XX, 95% CI XX to XX), which should be like (AOR 2.17, 95% CI 1.07 to 4.40), and make it consistent in the entire report. But it still has not been fixed in the abstract section. So make the consistency of your findings presentations, i.e., at (AOR 7.30, 95% CI 2.51 to 21.22), negative childhood experiences (AOR 7.11, 95% CI 3.56 to 14.20) to all abstract and FINDINGS sections for the consistency with neat and clean presentation.

Response: Thank you for your suggestion and we tried to correct it in the revised manuscript as follows:

Results: The prevalence of suicidal behavior was found to be 19.8% with 95% CI (16.6, 22.8). Regarding predictive variables: low social support (AOR 12.42, 95% CI 5.95 to 25.95), expressed more anger (AOR 7.30, 95% CI 2.51 to 21.22), negative childhood experiences (AOR 7.11, 95% CI 3.56 to 14.20), family history of suicidal attempts (AOR 5.66, 95% CI 2.44 to 13.10), have no a good relationship with parents (AOR 5.59, 95% CI 2.59 to 12.08), depression (AOR 5.32, 95% CI 2.64 to 10.70), stressful life events (AOR 2.91, 95% CI 1.45 to 4.91), and interpersonal stress (AOR 2.17, 95% CI 1.07 to 4.40) were significantly increased odds of suicidal behavior among respondents.

• Conclusion section: make the section very clear and concise.

Response: Thank you for your suggestion and we tried to make it more concise and clear on the revised manuscript as follows:

Conclusion: Suicidal behavior was prevalent in the study area, and factors such as depression, stressful life events, poor parental relationships, negative childhood experiences, high anger expression, low social support, and a family history of suicidal attempts were the significant contributors. Mental health interventions have to emphasize psychosocial support for adolescents, improve the parent-adolescent relationship by increasing parental knowledge about adolescents’ mental health, and integrate youth-friendly health services that foster mental well-being. Policymakers can also propose mandating mental health guidance in schools.

• Remove the corresponding author's phone number, i.e., Phone: +251-913-69-28-80.

Response: Thank you for the comment, the phone number from corresponding is removed and corrected as follows

*Correspondent Author: Abdulbasit Sherfa, E-mail: abdusherfa3@gmail.com

Findings section:

• All table titles should be short and clear as per the last feedback and made consistent, i.e., Table 1:, Table 2., and Table 3.

Response: Thank you for your suggestion and we have made the necessary correction on the revised manuscript as follows

Table 1: Socio-demographic characteristics of participants

Table 2: The frequency distribution of participants’ psychosocial and substance-related factors

Table 3: Bi-variable and multivariable logistic regression of suicidal behavior and associated factors among participants

• The title of Figure 1 should be short and clear, i.e., Figure 1: Distribution of suicidal behaviour by level of anger expression

Response: Thank you for your suggestion and we have made the necessary correction on the revised manuscript as follows

Figure 1: Percentage distribution of suicidal behavior by level of anger expression

• Table two's title should be shorter as per earlier feedback, and category variables need to be made consistent where category variables are inconsistent with other tables, i.e., the first letters are small letters, so make it consistent with other tables.

Response: Thank you for the comment; we have made the necessary correction on the revised manuscript as follows

Table 2: The frequency distribution of participant’s psychosocial and substance-related factors

Variable Category Frequency Percentage

Family history of suicide attempt Yes

No 85

546 13.5

86.5

Family history of suicidal committed Yes

No 2

629 0.3

99.7

Grow up with your biological family Yes

No 503

128 79.7

20.3

Relationship with parent

Good to very good

Not good/disturbed or very disturbed 542

89 85.9

14.1

Relationship with peers

Good

Poor 629

2 99.7

0.3

Social support

Poor

Moderate

Strong 251

300

80 39.8

47.5

12.7

Ever used tobacco/ cigarettes Yes

No 28

603 4.4

95.6

In the past three month have you used tobacco/cigarettes Yes

No 23

608 3.6

96.4

In your life, have you Ever used alcohol Yes

No 58

573 9.2

90.8

in the past three month have you used alcohol Yes

No 50

581 7.9

92.1

Ever use of khat

Yes

No 32

599 5.1

94.9

Khat use in the past three month Yes

No 26

605 4.1

95.9

• Figure 1 removes the table and provides the % value at the top of the bar diagram and removes the below tables and keeps the legend.

Response: Thank you for the comment; we have made the necessary correction on the revised manuscript as follows

Figure 1: Percentage distribution of suicidal behaviour by level of anger expression

• The discussion section should be thoroughly reviewed and succinctly presented without repeating all findings with 95% CI values, but you can use AOR only and compare and contrast with global and national empirical evidence and provide your thoughts on society.

Response: we have made necessary correction based on your suggestion in the revised manuscript as follow:

Discussion

The primary objective of this study was to evaluate the prevalence and factors influencing suicidal behavior among adolescents residing in Bahir Dar city. The finding indicated that the prevalence of suicidal behavior was 19.8% (95% CI: 16.6, 22.8). Key factors associated with suicidal behavior included a family history of suicide, exposure to stressful life events, adverse childhood experiences, poor social support, troubled parent-child relationships, depression, and high levels of anger expression. This study underscores that suicidal behavior is a significant public health concern for adolescents in Bahir Dar city.

Prevalence of suicidal behaviors

The present finding was in line with a study of high school students in Ethiopia that found that 22.5% had suicidal ideation and 16.2% had attempted suicide (32). Furthermore, the results were consistent with studies conducted in Ghana among high school students, which found that 18.2% had suicidal thoughts, 22.5% had suicidal plans, and 22.2% had attempted suicide (42); in Mozambique, where 17.7% had suicidal ideation and 18.5% had attempted suicide (43); and in Togo (16.5% had suicidal thoughts) (44).

However, the finding in this study was higher than the school-based study done in Kut City, which reported suicidal behavior to be 8.3% (45), and Tunisia (9.6% suicidal ideation and 7.3% suicidal attempt) (46). On the contrary, the finding was lower than studies done in Liberia, where the prevalence of suicidal attempts and suicidal ideation were 33.7% and 26.8% among adolescent students, respectively (47), Lebanese suicidal ideation was 28.9% (12), and Benin (23.2% suicidal thought and 28.3% suicidal attempt) (21). The disparity may be explained by differences in the sample size, study setting, socio-demographic, economic, and cultural characteristics of participants, as well as by the measurement tools used; the Mini International Neuropsychiatric Interview (MINI) Suicidal Scale was used in this study (33).

Factors associated with suicidal behavior

Suicidal behavior was 12.42 times more likely to occur in adolescents with poor social support than in those with good social support. Likewise, other studies conducted in Ethiopia showed a relationship between suicidal behavior and a lack of social support (2, 32, 48). Likewise, adolescents who did not have a good relationship with their parents were 5.59 times more highly at risk for suicidal behavior than those who had a good relationship with their parents. These findings were supported by prior research conducted in China and Japan (16, 50). These relationships may arise from the fact that being neglected by family and friends and not getting instrumental, informational, and emotional support from important others greatly increases feelings of worthlessness and hopelessness, which in turn contribute to suicidal behavior (49).

Adolescents who experienced negative childhood experiences (abuse, neglect, and household dysfunction) were seven times more likely to exhibit suicidal behavior than their peers. Previous research has revealed that adolescents who have experienced abuse, physical harm, or violence are more likely to engage in suicidal behavior (36). Adolescents who had to go through stressful events in their lives had a 2.91 times higher risk of suicidal behavior than their counterparts. This finding was supported by previous studies in which suicidal behavior was significantly favored by stressful life events (46, 51). Adverse childhood experiences and stressful life events can have long-term psychological consequences, increasing vulnerability to mental health disorders, including depression and anxiety, or intensifying pre-existing conditions, increasing suicide ideation and behavior (52-54).

Adolescents who had to go through difficult life situations had almost a threefold higher risk of suicidal behavior than their counterparts. Previous research has found that stressful life situations are much more likely to promote suicidal conduct (46, 51). Adverse childhood experiences and stressful life events can have long-term psychological consequences, including greater vulnerability to mental health concerns such as depression and anxiety, or they can trigger or worsen existing mental health difficulties, leading to an increase in suicidal thoughts and actions.

Adolescents having a family history of suicidal behavior were 5.66 times more likely to attempt suicide than those without such a history. This finding was supported by a study conducted in Fitche Town, North Shoa, Oromia region (48), while no significant relationships were discovered in Dangila town (32). The difference could be due to the study subjects' strategies for coping, sample size, and study setting.

Participants with depression had a five times higher risk of suicidal behavior than those without depression. Previous research has indicated that depression in adolescents has a positive association with suicidal behavior (22, 45, 55). Participants with high interpersonal stress had roughly twice the risk of suicidal behavior as those with low interpersonal stress. A previous study also supports that high interpersonal stress was significantly associated with suicidal behavior (56). A potential explanation for this is that depression and interpersonal stress have often been recognized as important correlations or predictors of suicide conduct in adolescents.

Finally, the current study revealed that adolescents with higher anger expression behavior were roughly seven times more likely to report suicidal behavior than those with lower anger expression behavior. Individuals with anger-related illnesses may demonstrate a variety of cognitive and interpersonal impairments that harm their well-being and contribute to a higher risk of suicidal conduct, such as poor social problem-solving ability and reasoning. The outcomes of this study offer support to the idea that high levels of external angry emotions are significantly associated with depressed symptoms and suicidal behavior.

• After completing all revisions, please fully proofread to make it clear and coherent.

Response: we tried to check the entire manuscript and made it consistent.

Response: The figure is uploaded to the Preflight Analysis and Conversion Engine (PACE) digital diagnostic tool, https://pacev2.apexcovantage.com/. PACE ensured that the figures meet PLOS requirements.

---

## [Editor Report · Decision Letter 3]

8 Apr 2025

Prevalence and associated factors of suicidal behavior among adolescents in Bahir Dar City, Northwest Ethiopia

PONE-D-23-31943R3

Dear Abdulbasit Sherfa,

I greatly appreciate all authors for thoroughly reviewing your manuscript and addressing all the suggestions, which have been addressed well. We’re pleased to inform you that your manuscript has been judged scientifically suitable for publication and will be formally accepted for publication once it meets all outstanding technical requirements. We appreciate the effort you have put into addressing the reviewers' comments during the revision process, which has greatly enhanced the quality of your manuscript. 

Within one week, you’ll receive an e-mail detailing the required amendments. When these have been addressed, you’ll receive a formal acceptance letter, and your manuscript will be scheduled for publication.

Thank you for choosing the Journal of PLOS ONE for disseminating your research. We look forward to seeing your work published and hope it will inspire further advancements in the field.

Kind regards,

Sharada P Wasti, MSc, PhD

Academic Editor

PLOS ONE

---

## [Editor Report · Acceptance letter]

PONE-D-23-31943R3

PLOS ONE

Dear Dr. Sherfa,

I'm pleased to inform you that your manuscript has been deemed suitable for publication in PLOS ONE. Congratulations! Your manuscript is now being handed over to our production team.

Kind regards,

on behalf of

Dr. Sharada P Wasti

Academic Editor

PLOS ONE